# Aflatoxin M_1_ Levels in Milk and Urine Samples of Nursing Mothers in Bangladesh: Exposure Assessment of Infants

**DOI:** 10.3390/toxins17080399

**Published:** 2025-08-08

**Authors:** Humaira Rashid Tuba, Sohel Rana, Khandaker Atkia Fariha, Gisela H. Degen, Nurshad Ali

**Affiliations:** 1Department of Biochemistry and Molecular Biology, Shahjalal University of Science and Technology, Sylhet 3114, Bangladesh; humairatuba39@gmail.com (H.R.T.); atkiafariha-bmb@sust.edu (K.A.F.); 2Department of Veterinary and Animal Science, Rajshahi University, Rajshahi 6205, Bangladesh; sranadvm@gmail.com; 3Leibniz-Research Centre for Working Environment and Human Factors (IfADo), TU Dortmund, Ardeystr. 67, D-44139 Dortmund, Germany; degen@ifado.de

**Keywords:** aflatoxin M_1_, Bangladesh, biomarker, breast milk, urine, contaminant, exposure, infants

## Abstract

Breast milk is the ideal source of nutrition for infant growth and development. However, when nursing mothers consume aflatoxin B_1_ (AFB_1_)-contaminated food, the hydroxylated form aflatoxin M_1_ (AFM_1_) is transferred to breast milk and urine. AFB_1_ and its metabolite AFM_1_ are potent carcinogens and can pose significant risks to food safety and public health worldwide. This study determined the prevalence of AFM_1_ in the breast milk and urine of nursing mothers in Bangladesh and estimated infant exposure to this toxin. Breast milk and urine samples (72 each), collected from nursing mothers in three different regions of the country, were analyzed for AFM_1_ occurrence via a sensitive enzyme-linked immunosorbent assay (ELISA). AFM_1_ was present in 88.9% of urine samples, with a mean concentration of 109.9 ± 52.8 pg/mL, ranging from 40.0 to 223.8 pg/mL. AFM_1_ was also detected in 50% of the breast milk samples, with a mean concentration of 4.6 ± 0.7 pg/mL, ranging from 4.0 to 6.1 pg/mL. A strong correlation (r = 0.72) was observed between milk and urinary AFM_1_ levels, indicating these as suitable biomarkers of AFB_1_ exposure. Yet, no significant correlations were identified between AFM_1_ levels in either milk or urine and the food items typically consumed by nursing mothers. The average estimated daily intake (EDI) for AFM_1_ with breast milk was 0.59 ng/kg bw/day, with no significant difference between infants (0.57 ng/kg bw/day) and toddlers (0.65 ng/kg bw/day). A comparison of computed EDI ranges for AFM_1_ with a proposed tolerable daily intake value resulted in Hazard Indices below 1 for all exposure scenarios. This indicates that the AFM_1_ concentrations in breast milk from three regions of Bangladesh raise no concern. Also, the measured levels were far lower than the maximal levels set in the EU regulations for AFM_1_ in dairy milk and infant formula (50 ng/kg and 25 ng/kg, respectively).

## 1. Introduction

Breast milk is the ideal source of nutrition for newborns and infants due to its unique composition, which encompasses essential nutrients, bioactive factors, and immune-enhancing components [1,2]. The World Health Organization recommends exclusive breastfeeding for the first six months of life, with a continued focus on breastfeeding until the child is two years old or even longer [3]. This underscores the vital role of breastfeeding in fostering optimal health and development during early childhood. However, mycotoxins such as aflatoxins can transfer to breast milk when nursing mothers ingest these toxins, thereby potentially exposing their infants [4,5].

Aflatoxins, a group of fungal toxins produced mainly by *Aspergillus flavus* and *Aspergillus parasiticus*, are important contaminants in feed and food [6,7]. They pose a significant public health risk due to severe toxic effects on animals and humans, best documented for the most potent mycotoxin aflatoxin B_1_ (AFB_1_) [8,9,10]. All aflatoxins are classified as human carcinogens based on sufficient evidence from both experimental studies and epidemiological data on liver cancer, applying biomarkers [11,12].

AFB_1_ metabolism primarily occurs in the liver and involves the action of phase I and phase II enzymes that convert AFB_1_ into several metabolites [13,14]. One key process is the bioactivation of AFB_1_ via cytochrome P450 enzymes to the DNA-reactive intermediate AB_1_-8,9-epoxide, which is detoxified via conjugation with glutathione. Additionally, AFB_1_ is hydroxylated to aflatoxin M_1_ (AFM_1_), a carcinogenic and hepatotoxic metabolite that can undergo bioactivation to a mutagenic epoxide. AFB_1_ is also converted to aflatoxin Q_1_ and other less toxic metabolites [14].

In humans, the ingestion of AFB_1_ will result in the excretion of AFM_1_ in breast milk and urine [15]. Mycotoxin exposure occurs mainly through the consumption of maize and its derivatives, nuts, dairy products, sunflower oil, bread, and whole grain cereals [8,16]. In children from low- and middle-income countries, aflatoxin exposure has been associated with growth retardation and weakened immune function [17,18]. Children are more susceptible to aflatoxins than adults because of their smaller body weight, decreased ability to detoxify these toxins, and reliance on cereal-based post-weaning foods [19,20].

To protect consumers, many countries have established regulatory standards for aflatoxins in food and feed [7]. Bangladesh also introduced regulations on aflatoxin contamination for specific food items, including groundnuts, almonds, Brazil nuts, hazelnuts, pistachios, and for AFM_1_ in dairy milk [21]. Yet, data on AFB_1_ and its metabolite AFM_1_ in food commodities in Bangladesh is scarce due to the absence of regular surveillance for these contaminants [9,22]. In such a situation, the analysis of biological samples (blood, breast milk, and urine) offers valuable insights into human mycotoxin exposure [18]. AFM_1_ is excreted in both urine and human milk, and is thus a valuable biomeasure of AFB_1_ exposure from food [23,24]. Our previous biomonitoring studies analyzed AFM_1_ in urine samples collected in several districts of Bangladesh from adults and young children [25,26,27]. The data revealed some regional and seasonal fluctuation in AFM_1_ levels, indicative of widespread yet variable dietary mycotoxin exposure in the population. A first study on AFM_1_ presence in human milk samples in the Sylhet region of Bangladesh then served to estimate the exposure of nursed infants [28]. The present study analyzed AFM_1_ occurrence in both urine and milk samples collected from mothers in three different regions of Bangladesh with the aim of further assessing maternal and infant exposure. Although AFB_1_ is more potent than its metabolite AFM_1_, they have similar toxic properties regarding carcinogenic and hepatotoxic effects and bioactivation to mutagenic epoxides [8,15].

## 2. Results

### 2.1. Characteristics of the Study Participants

The participants’ characteristics are summarized in Table 1. The average age of the participants was 25.4 ± 5.4 years, with a BMI of 21.9 ± 4.2 kg/m^2^. A small percentage of nursing mothers completed secondary (5.4%) and graduate-level education (9.5%); most had only completed elementary (39.2%) or secondary (40.5%) education. In terms of socioeconomic status, most participants belonged to a low (47.2%)- or medium (43.1%)-socioeconomic groups. Notably, the majority of the mothers exhibited limited knowledge of diet and nutrition. The infants’ and toddlers’ ages ranged from 15 days to 12 months and from 13 to 30 months, respectively. The average body weight was 6.6 ± 1.8 kg for infants and 9.3 ± 1.9 kg for toddlers. The average urinary creatinine concentration among the participants was 0.76 ± 0.54 mg/mL.

### 2.2. AFM_1_ in Milk and Urine Samples

The levels, detection frequency, and distribution of AFM_1_ determined in milk and urine samples are detailed in Table 2 and Figure 1. AFM_1_ was found in breast milk at an average concentration of 4.6 ± 0.7 pg/mL, with a range from 4 to 6.1 pg/mL. In urine samples, AFM_1_ was present at a mean concentration of 109.9 ± 52.8 ng/L, with a range from 40 to 223.8 pg/mL. The creatinine-adjusted mean AFM_1_ concentration in the urine of all mothers was 235.4 ± 260.7 pg/mg creatinine. There was no statistical significance in the average AFM_1_ levels for either milk or urine samples taken from the three regions. The co-occurrence of AFM_1_ in both milk and urine samples was 47.2%. There was a strong and significant correlation (r = 0.719, *p* = 0.001) between the AFM_1_ concentration in milk and in urine, as shown in Figure 1. This supports the view that biomarker measurements in both matrices reflect maternal dietary exposure to AFB_1_ well.

### 2.3. Estimation of AFM_1_ Intake with Breast Milk and Risk Assessment Approach

The average estimated daily intake (EDI) for AFM_1_ was 0.59 ± 0.25 ng/kg bw/day, with a range from 0.30 to 1.12 ng/kg bw/day. Infants had an EDI of 0.57 ± 0.25 ng/kg bw/day (range: 0.30–1.04 ng/kg bw/day), while toddlers had a higher EDI of 0.65 ± 0.26 ng/kg bw/day (range: 0.37–1.12 ng/kg bw/day) (Table 3). Among infants, those aged 1–2 months had a slightly higher mean AFM_1_ intake of 0.61 ± 0.25 ng/kg bw/day compared to other infant age groups, but still lower than the intake for toddlers (0.65 ± 0.26 ng/kg bw/day). Nevertheless, the differences in the EDI were not statistically significant. It is worth noting that an EDI for toddlers may not be a reliable estimate since children >13 months typically consume breast milk only occasionally, primarily relying on post-weaning foods that could have higher levels of contaminants than human milk.

To assess the EDI values for AFM_1_ in nursed children in terms of risk, one can compare them with the TDI value of 2 ng/kg bw/day proposed by Kuiper-Goodman (1994) [29], an approach also applied previously (see Section 5 on AFM_1_ exposure and risk assessment approach for more details). Table 3 lists the outcomes for all infant age groups and toddlers and for several exposure scenarios: the calculated mean Hazard Index (HI) for AFM_1_ was 0.27 (with a median of 0.19 and a maximum of 0.52) in infants and 0.32 (with a median of 0.37 and a maximum of 0.56) in toddlers. The 95% confidence intervals for the lower and upper bounds were 0.25 and 0.32, respectively, in infants, and 0.26 and 0.39, respectively, in the toddler group. Overall, the outcomes—with all HI values < 1—do not raise a concern. Further aspects related to potential risks of AFM_1_ exposure in infants through breast milk are considered in the Section 3.

### 2.4. Correlation Between AFM_1_ Biomarker Levels and Food Consumption

A potential correlation between the dietary habits of mothers and their AFM_1_ levels in milk and urine samples was examined. We applied Spearman’s correlation analysis and focused on key food items, such as rice, wheat/maize, milk and dairy products, and groundnuts (see Table 4). Significant correlations were not detected between AFM_1_ concentrations in breast milk samples and the consumption frequency of rice (*p* = 0.192), wheat/maize (*p* = 0.449), milk and dairy products (*p* = 0.516), or groundnuts (*p* = 0.724). Likewise, no noteworthy correlation was found between urinary AFM_1_ levels and the intake of any of the aforementioned food items.

## 3. Discussion

This biomonitoring study provides new insights on the aflatoxin exposure of nursing women in Bangladesh, their breastfed infants, and the potential risks associated with AFM_1_ in human milk. Analysis of urine from our cohort found AFM_1_ in 89% of the samples at a mean concentration of 110 ± 53 pg/mL (range 40–224 pg/mL, Table 2), related to dietary AFB_1_ exposure in mothers from three different districts in Bangladesh. The new urinary AFM_1_ biomarker data resemble the findings of a recent study on workers of grain mills and persons with no occupational contact with crops: in both groups, AFM_1_ was found in most of their urine (96.1% and 92%), at similar mean concentrations (106.5 ± 35.0 pg/mL 123.3 ± 52.4 pg/mL) and ranges [30]. An early study on rural and urban residents of the Rajshahi district of Bangladesh detected AFM_1_ in 46% of urine samples at 80 ± 60 pg/mL in a range from 31 to 348 pg/mL [25]. Two additional studies reported similar or lower levels of AFM_1_ in the urine of adults, pregnant women, and children [26,27]. Overall, these data suggest widespread exposure to dietary AFB_1_, along with some regional differences in the frequency of detection and levels of the AFM_1_ biomarker. These variations may be related to differences in dietary practices and/or the mycotoxin contamination of foods consumed by the Bangladeshi population.

Along with the presence of biomarkers in the urine of nursing women, AFM_1_ was also detected in 50% of human milk samples at 4.6 ± 0.7 pg/mL (range from 4 to 6.05 pg/mL), i.e., at concentrations clearly lower than those measured in urine. This finding is in line with early studies reporting that a higher fraction of ingested AFB_1_ (1 to 3%) occurs as AFM_1_ in urine [23] than that (0.1 to 0.43%) transferred to breast milk [24]. In the present study, a strong correlation (r = 0.72) was observed between milk and urinary AFM_1_ levels (Figure 1), suggesting its analysis in both matrices as a useful biomarker of exposure.

The level and detection frequency of AFM_1_ in the present study closely resemble those in our previous analysis, which detected AFM_1_ in 51.6% of 62 breast milk samples, with a range from 4 to 6.66 pg/mL and a mean level of 4.42 ± 0.56 pg/mL [28]. Our data on AFM_1_ in human breast milk in Bangladesh can be compared to findings from other countries: for instance, mean concentrations of 17 pg/mL in Mexico [31], 74.4 pg/mL in Egypt [32], and 401 pg/mL in Sudan [33]. These levels are far higher than those found in breast milk from our Sylhet cohort of nursing women. AFM_1_ prevalence and levels in breast milk can vary considerably between countries, regions, and seasons [15,28]. This has been related to the AFB_1_ contamination of local foods and to maternal dietary habits.

In our cohort, we looked for a correlation between the AFM_1_ biomarker level and the frequency of consuming staple food such as rice, wheat/maize, and other items (groundnuts and milk and dairy products). The analysis (Table 4) did not reveal any significant correlations with the consumed food items, which may be attributed to rather low AFM_1_ concentrations detected in Bangladeshi mothers. In similar studies involving cohorts of women with higher AFM_1_ levels in breast milk, stronger correlations with their consumption of specific food groups, such as egg, cola drink, sunflower oil, bread, bakery products, and cured pork meat, have been observed [34,35].

Data on the presence of aflatoxins and other mycotoxins in food commodities in Bangladesh is limited [26,36]. One study on AFB_1_ analysis in rice, lentils, wheat flour, dates, betelnut, red chili powder, ginger, and groundnuts reported that five out of the eight commodities surpassed EU regulatory limits, with the highest concentrations detected in dates and groundnuts [37]. Another study analyzed total aflatoxins (AFB_1_, AFB_2_, AFG_1_, and AFG_2_) in cereal crops collected in six districts over one year. It found that maize had the highest levels of aflatoxins, while rice and wheat showed lower contamination, with all having considerable seasonal variability [38]. Two more recent studies have measured AFM_1_ contamination in raw and processed dairy milk. Tarannum et al. [39] analyzed samples collected in the Dhaka district and reported higher mean concentrations of AFM_1_ in raw milk (699 ng/L) than in processed milk (pasteurized and UHT milk with 100 and 36 ng/L, respectively). About 75% of all positive samples were above the EU regulatory limit for AFM_1_. Also, Sumon et al. [40] found that 78.6% of all milk samples contained AFM_1_, ranging from 5.0 to 198.7 ng/L. About one-third of raw, pasteurized, and UHT milk samples exceeded the EU limit of 50 ng/L for AFM_1_. Notably, the levels of AFM_1_ in dairy milk were considerably higher than those found in human breast milk, indicating that dairy milk is less safe for infants than their mother’s milk.

Regarding the exposure of infants in Bangladesh, AFM_1_ concentrations detected in breast milk were used to determine the EDI values for infants across different age categories. The AFM_1_ intake ranged between 0.3 and 1.04 ng/kg bw/day, with a mean of 0.57 ng/kg bw/day for all infants, values slightly lower than those for toddlers (Table 3). To assess risks related to AFM_1_ exposure, we chose an approach used before by others comparing the EDI values to a TDI value of 2 ng/kg bw/day (see Section 5). The EDI scenarios (95th CI for means) for infants or toddlers did not exceed this TDI value, and expressed as the HI, all ratios were below 1 (Table 3). This assessment indicates that the low AFM_1_ exposure in breastfed infants does not raise concern. Furthermore, AFM_1_ levels in breast milk samples from Bangladesh in this study and our previous one [28] are lower than the strict limit of 25 ng/kg established for AFM_1_ in infant formula [41].

Higher EDI values for AFM_1_ in breastfed infants have been reported in some countries. However, switching to infant formula is not a feasible alternative in developing countries with financial constraints [42]. Given its well-known benefits, breastfeeding should be promoted rather than discouraged, and measures should be implemented to enhance food safety and minimize human exposure to mycotoxins.

Our study has several limitations. Firstly, while it includes three different regions of Bangladesh, the participant count was fairly low, and our results may not reflect the scenario in Bangladesh completely. Secondly, we collected milk and urine samples just once, which fails to examine seasonal variations in AFM_1_ biomarker levels. Third, ELISA is a rapid and sensitive method, but focused on measuring AFM_1_ in breast milk and urine. Future studies that apply sensitive LC-MS/MS multi-biomarker methods for the analysis of milk and urine samples could enhance our understanding of the concurrent exposure to various mycotoxins of nursing mothers in Bangladesh and their infants.

## 4. Conclusions

In conclusion, AFM_1_ was detected in half of the milk samples (range 4.0 to 6.1 pg/mL) and nearly ninety percent of the urine samples (range 40.0 to 223.8 pg/mL) from nursing women in Bangladesh, indicative of their dietary exposure to AFB_1_. The AFM_1_ concentrations in human milk were found to be rather low and result in AFM_1_ intake values for breastfed infants that do not raise concern. However, this study is based on a limited number of participants and only one sampling period. Therefore, further research involving a larger cohort from different regions of the country, alongside the application of sensitive and targeted biomarker analysis methods, is recommended to gather more comprehensive insights into maternal exposure as well as infant exposure through breast milk in Bangladesh.

## 5. Materials and Methods

### 5.1. Study Participants and Areas

This cross-sectional study was carried out between March 2021 and November 2021, with all laboratory analysis conducted at the Department of Biochemistry and Molecular Biology at Shahjalal University of Science and Technology in Sylhet, Bangladesh. A total of 72 nursing mothers from three regions—Sylhet, Cumilla, and Mymensingh—took part in the study. Women who faced any acute or chronic illnesses during their pregnancy were not included. The study focused on healthy mothers and their breastfed offspring, which comprised 53 infants aged 12 months or younger and 19 toddlers aged 13 to 30 months. Mothers completed a brief questionnaire that gathered information on anthropometrics, socio-demographics, economic status, and general dietary habits. Written consent was obtained from participants prior to their inclusion in the study. The Internal Ethics Review Board at the Department of Biochemistry and Molecular Biology, School of Life Sciences, Shahjalal University of Science and Technology in Sylhet, Bangladesh approved the research (Reference number: 01/BMB/2020).

### 5.2. Food Consumption Information

The nursing mothers were asked to fill out a short food frequency questionnaire (FFQ) about their usual eating habits. This FFQ featured food items frequently consumed in Bangladesh, including rice, wheat, maize, groundnuts, and a variety of milk and dairy products. Among these items, rice was consumed regularly by most of the participants, with some reporting intake up to three times a day. The frequency of food consumption was assessed on a scale of 1 to 5 (see Table 4) and as described before [43].

### 5.3. Sample Collection and Preparation

Approximately 10–15 mL of both urine and breast milk samples was collected from every participant in sterile tubes. Although we supplied breast pumps for the mothers, the majority chose to express their milk manually. The urine samples collected were spot samples. These urine samples were then transported in an icebox and stored at −20 °C in the laboratory for AFM_1_ analysis. The urine’s creatinine content was quantified using a colorimetric method following the manufacturer’s guidelines (HUMAN Gesellschaft für Biochemica und Diagnostica mbH, Wiesbaden, Germany) and a semi-automatic biochemistry analyzer (Humalyzer 3000, Medicon Services, Tuttlingen, Germany). AFM_1_ concentrations in both breast milk and urine were assessed through a competitive enzyme-linked immunosorbent assay (ELISA) using a commercial kit from Helica Biosystems Inc. (Santa Ana, CA, USA; catalog no. 961AFLM01M-96 for milk and Cat. No. 991AFLM01U-96 for urine), following the manufacturer’s instructions and previously established methods [25,28]. In brief, the milk samples were refrigerated overnight to promote the coagulation of fat molecules. Subsequently, samples were centrifuged at 2500× *g* for 10 min at RT to enable the separation of the upper fatty layer and facilitate its removal via aspiration; the remaining lower layer of skimmed milk was utilized directly for the assay. Urine samples were centrifuged at 3030× *g* for 5 min, and the supernatant was then used for AFM_1_ determination according to the specified protocol.

### 5.4. Determination of AFM1 in Milk and Urine Samples

The concentration of AFM_1_ in milk samples was measured as described elsewhere [28]. Briefly, 200 μL of standard and test samples was added in duplicate to pre-coated plates. After 2 h of incubation at RT, the wells were emptied and washed three times with washing buffer. Then, 100 μL of conjugate was added and incubated for 15 min. Following another wash, 100 μL of enzyme substrate was added and incubated for another 15 min. Finally, 100 μL of stop solution was added, mixed gently, and absorbance at 450 nm was measured with an ELISA plate reader (Apollo 11 LB 913, Berthold, Germany) within 15 min.

As described before [25], the analysis of AFM_1_ in urine involves diluting AFM_1_ standards and urine samples with distilled water at ratios of 1:20 or 1:5. A volume of 100 μL from each sample is mixed with 200 μL of assay buffer and transferred to antibody-coated microtiter wells, where it is incubated at room temperature for 1 h. Following incubation, the plate is washed three times with the provided washing solution. Subsequently, 100 μL of AFM_1_ conjugate is added to each well and incubated for an additional 15 min. The plate is then washed again to eliminate any unbound conjugate. After washing, 100 μL of substrate reagent is added for the color reaction to develop for 15 min in the dark. Then, 100 μL of stop solution is added, and the absorbance is measured at 450 nm with a microplate reader (Apollo 11 LB 913, Berthold, Germany). The absorption intensity is inversely related to the AFM_1_ concentration in the samples. The AFM_1_ levels were determined using the standard curves for the AFM_1_ solutions of each plate.

All milk and urine samples were analyzed in duplicate, with the average values calculated. Recovery studies used spiking skimmed milk samples with AFM_1_ concentrations of 5 pg/mL and 10 pg/mL, while urine samples were spiked with AFM_1_ concentrations of 20 and 50 pg/mL. The repeatability showed acceptable precision, with relative standard deviations (RSDs) below 5% for milk and 6% for urine. Recovery rates were 97–102% for milk and 88–120% for urine. The method detection limit (MDL) was 4 pg/mL for milk and 40 pg/mL for urine.

### 5.5. AFM_1_ Exposure and Risk Assessment Approach

The daily intake estimate (EDI) of AFM_1_ for breastfeeding infants was determined using the following equation:

EDI_AFM1_ (ng/kg bw/day) = (V_BM_ × C_AFM1_)/BW, where V_BM_ represents the typical volume of milk an infant consumes daily (L), C_AFM1_ is the concentration of AFM_1_ (ng/L), and BW denotes the baby’s body weight (kg). For infants up to 2 months old, the average daily milk consumption is around 150 mL; for those aged 2 to 4 months and older, the average intake is estimated to be 185 mL per day, according to EFSA [44]. The body weights used were those of the individual infants. The risks associated with the ingestion of AFM_1_ through breast milk were evaluated by comparing the EDI results to a tolerable daily intake (TDI) of 2 ng/kg bw/day, a value proposed before [29] and applied in previous studies on AFM_1_ exposure from milk [29,45,46]. The hazard quotient (HI, or Hazard Index, which is also referred to as the %TDI) was computed using the following formula: HI = EDI/TDI.

### 5.6. Statistical Analysis

Statistical data analysis was conducted using IBM SPSS Statistics version 26.0. The results were expressed as percentages (%), means, medians, and percentiles. An independent-sample *t*-test was performed to compare the EDI of AFM_1_ between the groups of infants and toddlers. To determine the correlation between maternal milk AFM_1_ levels and urinary AFM_1_ levels, Spearman’s correlation coefficient test was applied (two-tailed). Spearman’s correlation coefficient test (two-tailed) was also used to analyze the relationships between AFM1 levels in breast milk and urine samples with the frequency of food consumption data. A *p*-value of less than 0.05 was deemed statistically significant.

## Figures and Tables

**Figure 1 toxins-17-00399-f001:**
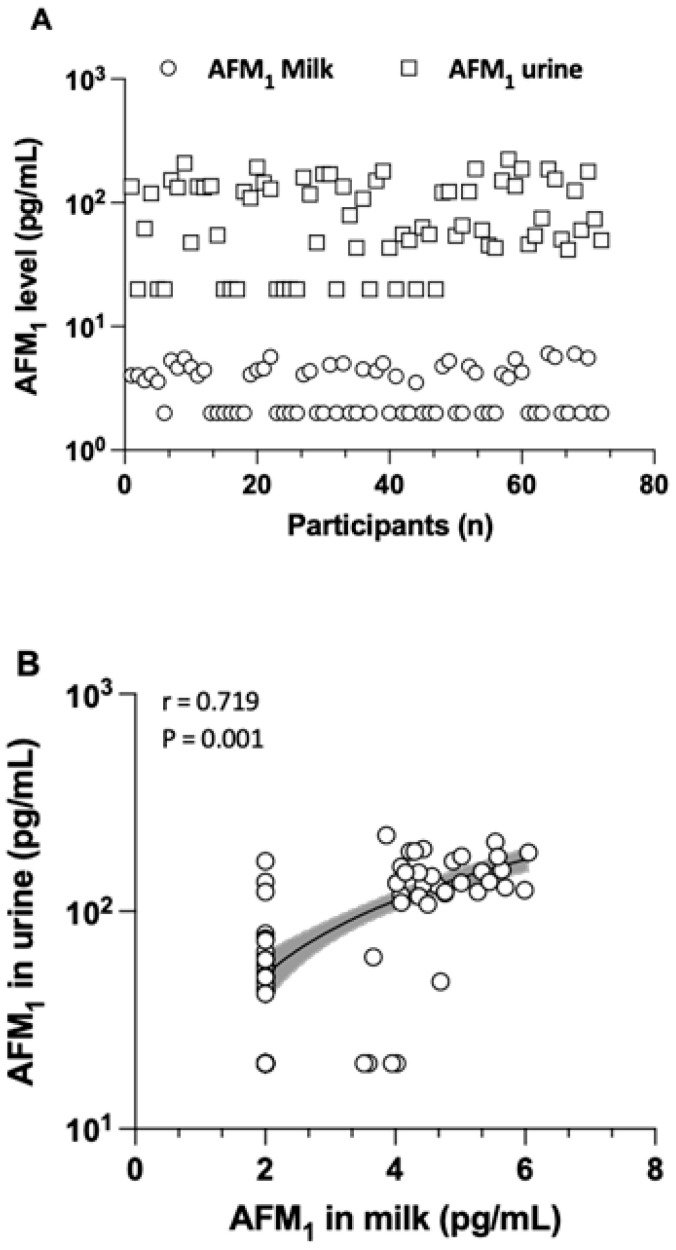
The distribution of AFM_1_ in both milk and urine samples (**A**) and the correlation between their levels in urine and milk (**B**), depicted with a logarithmic scale on the *Y*-axis. Samples with AFM_1_ levels below the MDL were considered as half of the MDL for the analysis of distribution and correlation. Only the samples that tested positive (≥MDL) were included to assess the co-occurrence of AFM_1_ in both milk and urine samples.

**Table 1 toxins-17-00399-t001:** Characteristics of the nursing mothers and their children.

Variables	Values
Nursing women (N)	72
Age (yrs)	25.4 ± 5.4
BMI (kg/m^2^)	21.9 ± 4.2
Creatinine in urine (mg/mL)	0.76 ± 0.54
Level of education (%)	
Illiterate	5.4
Primary	39.2
Secondary	40.5
Higher-secondary	5.4
Graduate and above graduate	9.5
Occupation (%)	
Students	2.7
Housewives	91.9
Office workers	4.0
Others	1.4
Occupation of spouse (%)	
Farmer	8.1
Office workers	21.6
Business	28.4
Others	41.9
Socioeconomic status (%)	
Low	47.2
Medium	43.1
Upper medium	9.7
Knowledge of diet and nutrition (%)	
Low	68.7
Medium	25.4
High	6.0
Children age (month)	
Infants (15 d–12 m)	6.0 ± 3.7
15–30 d	0.9 ± 0.2
31 d–2 m	1.8 ± 0.3
>2–<4 m	2.9 ± 0.3
4–12 m	7.9 ± 2.8
Toddlers (13–30 m)	19 ± 5.0 *
Childrens’ weight (kg)	
Infants (15 d–12 m)	6.6 ± 1.8
15–30 d	3.7 ± 0.9
31 d–2 m	4.3 ± 1.0
>2–<4 m	5.7 ± 0.8
4–12 m	7.5 ± 1.3
Toddlers (13–30 m)	9.3 ± 1.9 *

D: day, m: month. Data is shown as mean ± standard deviation or (%). * *p* < 0.001 in comparison to the infant cohort. *p* values were derived from the independent-sample *t* test.

**Table 2 toxins-17-00399-t002:** Levels of AFM_1_ in milk and urine samples of nursing mothers.

Sample Type	*N*	Positive Samples, *n* (%)	Mean ± SD, pg/mL	Median, pg/mL	Range, pg/mL
Milk	72	36 (50.0)	4.6 ± 0.7	4.5	4.0–6.1
Urine	72	64 (88.9)	109.9 ± 52.8 (235.4 ± 260.7) ^a^	121.2	40.0–223.8

Positive samples with the analyte present ≥ method detection limit (MDL: 4 pg/mL for milk and 40 pg/mL for urine). The mean and median values were computed using only positive samples. ^a^ Creatinine-adjusted AFM_1_ mean level (pg/mg creatinine).

**Table 3 toxins-17-00399-t003:** Estimated daily intake (EDI) of AFM_1_ (ng/kg bw/day) for infants and toddlers.

Group	N	Range	Median	Mean ± SD	95% Cl for Mean		HI (% TDI)				
					Lower Bound	Upper Bound	Mean	Median	Max	95% Cl for Mean	
										Lower Bound	Upper Bound
Infants (all)	53	0.30–1.04	0.37	0.57 ± 0.25	0.51	0.64	0.29 (29)	0.19 (19)	0.52 (52)	0.25 (25)	0.32 (32)
15–30 d	5	0.30–0.84	0.30	0.51 ± 0.28	0.15	0.86					
31 d–2 m	6	0.30–0.90	0.30	0.61 ± 0.25	0.34	0.88					
>2–<4 m	7	0.37–1.04	0.37	0.59 ± 0.30	0.31	0.87					
4–12 m	35	0.37–1.03	0.37	0.58 ± 0.24	0.49	0.66					
Toddlers (all)	19	0.37–1.12	0.74	0.65 ± 0.26	0.52	0.77	0.32 (32)	0.37 (37)	0.56 (56)	0.26 (26)	0.39 (39)
Total	72	0.30–1.12	0.49	0.59 ± 0.25	0.54	0.65	0.30 (30)	0.24 (24)	0.56 (56)	0.27 (27)	0.32 (32)

D: day, m: month. Infants are defined as those aged 12 months or younger, while toddlers are those aged between 13 and 30 months. Milk samples with AFM_1_ levels below the MDL of 4 ng/L were assigned half of the MDL for the purpose of calculating the EDI. Tolerable daily intake (TDI) of 2 ng/kg bw/day was used in the Hazard Index (HI) calculation (see the Section 5).

**Table 4 toxins-17-00399-t004:** Correlation between AFM_1_ levels and typical food consumption.

Food Items	AFM_1_ in Milk		AFM_1_ in Urine	
	Correlation (*r*)	*p*-Value	Correlation (*r*)	*p*-Value
Rice	−0.155	0.192	0.215	0.070
Bread (wheat/maize)	0.090	0.449	−0.045	0.709
Milk and milk products	0.059	0.516	−0.209	0.085
Groundnut	0.042	0.724	−0.172	0.150

Evaluation of food consumption frequency was conducted using numerical scores for major food items. For rice: 1 = 1 time per day, 2 = 2 times per day, and 3 = 3 times per day; for wheat/maize: 1 = 1 time per day, 2 = 2–3 times per week, 3 = not consumed at all, and 4 = on a weekly basis; for milk and milk-based products: 1 = daily, 2 = 2–3 times weekly, 3 = monthly, 4 = weekly, and 5 = not consumed at all; for groundnut: 1 = 1 time per day, 2 = monthly, 3 = not consumed, and 4 = weekly. Samples with positive biomarker results (≥MDL) were taken into account for the correlation analysis. *p*-values were calculated using Spearman’s correlation coefficient (two-tailed).

## Data Availability

The original contributions presented in this study are included in the article. Further inquiries can be directed to the corresponding author.

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
