# Peer review of "Aflatoxin M1 Levels in Milk and Urine Samples of Nursing Mothers in Bangladesh: Exposure Assessment of Infants"

_toxins, 2025, doi:10.3390/toxins17080399_

Round 1
Reviewer 1 Report
Comments and Suggestions for Authors
Breast milk is the ideal source of nutrition for infant growth and development. When nursing mothers consume AFB1-contaminated food, its metabolite AFM1 is transferred to breast milk. Therefore, it is important for exposure assessment of AFM1 levels in milk and urine samples of nursing mothers. This study determined the prevalence of AFM1 in breast milk and urine of nursing mothers in Bangladesh and estimated infant exposure to the toxin. The results will provide essential information about the intake levels of AFM1 for breastfed infants.
However, the manuscript contained some errors and questions that the authors should explain before it can be accepted.
Page 2 Line 76, The format should be consistent with the template, without spaces like line 35, line 43, etc.
The unit format should be the same throughout the text. For example, Line 85 “mg/ml”, but line 94, “pg/mL”, ml should be revised mL.
Line 136 “ng/kg b.w./day”, line 126 “ng/kg BW/day” and line 320 “ng/kg body weight/day”, the format should be the same.
line 289 to line 316, the authors described the determination method of AFM1 in the collected samples. However, the chromatographic condition needs to be supplemented.
Paragraph spacing should be modified to be consistent with the template.
Author Response
Response to Reviewer 1
Breast milk is the ideal source of nutrition for infant growth and development. When nursing mothers consume AFB1-contaminated food, its metabolite AFM1 is transferred to breast milk. Therefore, it is important for exposure assessment of AFM1 levels in milk and urine samples of nursing mothers. This study determined the prevalence of AFM1 in breast milk and urine of nursing mothers in Bangladesh and estimated infant exposure to the toxin. The results will provide essential information about the intake levels of AFM1 for breastfed infants. However, the manuscript contained some errors and questions that the authors should explain before it can be accepted.
Response: Many thanks for evaluating our manuscript and for providing comments and suggestions for further improvement.
Page 2 Line 76, The format should be consistent with the template, without spaces like line 35, line 43, etc.
Response: We agree, and the paragraphs spaces have been checked.
The unit format should be the same throughout the text. For example, Line 85 “mg/ml”, but line 94, “pg/mL”, ml should be revised mL.
Response: Thank you for the remark. The unit format has now been corrected.
Line 136 “ng/kg b.w./day”, line 126 “ng/kg BW/day” and line 320 “ng/kg body weight/day”, the format should be the same.
Response: This has now been corrected according to the suggestion and ng/kg bw/day has been applied throughout the entire text.
Line 289 to line 316, the authors described the determination method of AFM1 in the collected samples. However, the chromatographic condition needs to be supplemented. Paragraph spacing should be modified to be consistent with the template.
Response: We assume the question is due to a misunderstanding: the AFM1 levels in breast milk and urine samples were determined using the ELISA technique with no any chromatographic methods applied. We checked all paragraph spacing to ensure consistency.
Reviewer 2 Report
Comments and Suggestions for Authors
The article can have a significant interest to the readers as it is directly related to the infant growth and development. The study demonstrated that the nursing mothers consume aflatoxin B1 (AFB1) contaminated food, and that is transferred to breast milk and urine. AFB1 its metabolite AFM1 are potent carcinogens and can pose significant risks to food safety and public health worldwide. This study determined the prevalence of AFM1 in breast milk and urine of mothers in Bangladesh. The study is quite important to scientific community and need to include the following points for further improvement.
- What is the source of these toxins in food contamination?
- How did the that toxin metabolized? Are all metabolites of the the toxin are actually toxic or some are non-toxic too.
- Did author collect the data from hospitals for assessing the toxicity of this in infants.
- Mention the level of hazardous in the conclusion section.
Author Response
The article can have a significant interest to the readers as it is directly related to the infant growth and development. The study demonstrated that the nursing mothers consume aflatoxin B1 (AFB1) contaminated food, and that is transferred to breast milk and urine. AFB1 its metabolite AFM1 are potent carcinogens and can pose significant risks to food safety and public health worldwide. This study determined the prevalence of AFM1 in breast milk and urine of mothers in Bangladesh. The study is quite important to scientific community and need to include the following points for further improvement.
Response: We appreciate your time and effort in reviewing our work.
- What is the source of these toxins in food contamination?
Response: Aflatoxin B1 (AFB1) is mainly produced by certain fungi, notably Aspergillus flavus and Aspergillus parasiticus which can contaminate various types of foods (e.g. maize and nuts). This information is already mentioned in the 2nd paragraph of the introduction section.
- How did the that toxin metabolized? Are all metabolites of the toxin are actually toxic or some are non-toxic too.
Response: AFB1 metabolism primarily occurs in the liver, involving phase I and phase II enzymes that convert it into different metabolites. AFB1 undergoes bioactivation via cytochrome P450 enzymes (such as CYP1A2 and CYP3A4) which convert AFB1 into the reactive intermediate, aflatoxin B1-8,9-epoxide. The epoxide can be detoxified by conjugation with glutathione (GSH) via glutathione S-transferases, forming aflatoxin B1-GSH conjugates. AFB1 can be hydroxylated to produce aflatoxin M1, which is also carcinogenic and hepatoxic and undergo bioactivation to a mutagenic epoxide. AFB1 can also be transformed to aflatoxin Q1 and other less toxic metabolites. A brief information on AFB1 metabolism has now been included in the introduction section.
- Did author collect the data from hospitals for assessing the toxicity of this in infants.
Response: We collected samples from nursing mothers and gathered demographic information about their infants from urban, suburban, and rural areas across three different districts in Bangladesh. However, we did not collect any data from hospitals.
- Mention the level of hazardous in the conclusion section.
Response: The range of AFM1 concentration has been included in the conclusion section.
Not sure whether it is wise to mention the level again in the conclusion section
Reviewer 3 Report
Comments and Suggestions for Authors
Data such as those presented in this manuscript are important for monitoring public health. Overall, the study is well designed and clearly reported. The English is of high quality. The Conclusions are thoroughly explained and the reasoning behind them is clearly explained.
The "XXX" on line 127 does not appear to correlate to anything in the Methods section.
Author Response
Data such as those presented in this manuscript are important for monitoring public health. Overall, the study is well designed and clearly reported. The English is of high quality. The Conclusions are thoroughly explained and the reasoning behind them is clearly explained.
The "XXX" on line 127 does not appear to correlate to anything in the Methods section.
Response: Thank you for the appreciative comment on our study. Mentioning "XXX" in the method section was an error. It has now been removed and corrected accordingly.
Reviewer 4 Report
Comments and Suggestions for Authors
The aim of the manuscript entitled: ‘’Aflatoxin M1 levels in milk and urine samples of nursing mothers in Bangladesh: exposure assessment of newborns’’ was to investigate the AFM1 occurrence in both urines and milk samples collected from mothers in three different regions of Bangladesh with the aim to further assess maternal and infant exposure.
It is a well written manuscript with valuable biomonitoring results, presented in an understood manner. Please see few comments below.
Comments
Abstract: line 15, please provide the values (concentration levels) into one decimal point as in the table 2.
Line 127. It states ‘’ (see Methods section XXX for more details).’’ Please correct.
Line 182. Delete ‘’… n=….’’
Lines 194-196. It is reported ‘’ In similar studies involving cohorts of women with higher AFM1levels in breast milk, stronger correlations with their consumption of specific food groups have been observed’’ Please clarify which groups of food these are?
Author Response
Response to Reviewer 4
The aim of the manuscript entitled: ‘’Aflatoxin M1 levels in milk and urine samples of nursing mothers in Bangladesh: exposure assessment of newborns’’ was to investigate the AFM1 occurrence in both urines and milk samples collected from mothers in three different regions of Bangladesh with the aim to further assess maternal and infant exposure.
It is a well written manuscript with valuable biomonitoring results, presented in an understood manner. Please see few comments below.
Response: Thank you very much for your kind words and positive feedback on our manuscript.
Comments
Abstract: line 15, please provide the values (concentration levels) into one decimal point as in the table 2.
Response: The values into one single decimal point has been provided according to the suggestion.
Line 127. It states ‘’ (see Methods section XXX for more details).’’ Please correct.
Response: Thank you for mentioning the error. It has now been corrected.
Line 182. Delete ‘’… n=….’’
Response: It has been deleted.
Lines 194-196. It is reported ‘’ In similar studies involving cohorts of women with higher AFM1levels in breast milk, stronger correlations with their consumption of specific food groups have been observed’’ Please clarify which groups of food these are?
Response: A strong correlation has been observed between AFM1 in breast milk and consumption of specific food such egg, cola drink and sunflower oil, bread, bakery products and cured pork meat. This information has now been included.
Round 2
Reviewer 1 Report
Comments and Suggestions for Authors
Accept in present form.